# GraphAdapter: Tuning Vision-Language Models With Dual Knowledge Graph

**Xin Li**[1,2,*] **Dongze Lian**[2,*] **Zhihe Lu**[2,*] **Jiawang Bai**[3,2] **Zhibo Chen**[1,†] **and Xinchao Wang**[2,†]

[1] University of Science and Technology of China
[2] National University of Singapore
[3] Tsinghua University

lixin666@mail.ustc.edu.cn, {dongze,zhihelu}@nus.edu.sg, bjw19@mails.tsinghua.edu.cn
chenzhibo@ustc.edu.cn, xinchao@nus.edu.sg

## Abstract

Adapter-style efficient transfer learning (ETL) has shown excellent performance in the tuning of vision-language models (VLMs) under the low-data regime, where only a few additional parameters are introduced to excavate the task-specific knowledge based on the general and powerful representation of VLMs. However, most adapter-style works face two limitations: (i) modeling task-specific knowledge with a single modality only; and (ii) overlooking the exploitation of the inter-class relationships in downstream tasks, thereby leading to sub-optimal solutions. To mitigate that, we propose an effective adapter-style tuning strategy, dubbed GraphAdapter, which performs the textual adapter by explicitly modeling the dual-modality structure knowledge (*i.e.*, the correlation of different semantics/classes in textual and visual modalities) with a dual knowledge graph. In particular, the dual knowledge graph is established with two sub-graphs, *i.e.*, a textual knowledge sub-graph, and a visual knowledge sub-graph, where the nodes and edges represent the semantics/classes and their correlations in two modalities, respectively. This enables the textual feature of each prompt to leverage the task-specific structure knowledge from both textual and visual modalities, yielding a more effective classifier for downstream tasks. Extensive experimental results on 11 benchmark datasets reveal that our GraphAdapter significantly outperforms previous adapter-based methods. The code will be released at `https://github.com/lixinustc/GraphAdapter`

## 1 Introduction

Recent large-scale vision-language models (VLMs), such as CLIP [55] and ALIGN [27] have shown their promising representation capability for a series of downstream vision tasks [83], such as classification [44, 85, 42, 18, 86], generation [61, 31, 32, 60], and recognition [71, 73]. Different from previous pre-training models with ImageNet [20, 41, 13], based on large-scale parameters, VLMs can learn more prosperous, fine-grained, and consistent semantics from billions of text-image pairs with contrastive learning. This enables the VLMs with two advantages: 1) performing superior zero-shot generalization capability [89] with simple hand-craft prompts like "a photo of a [class]", and 2) possessing impressive transferability for downstream tasks [67, 69, 57, 65, 86] by finetuning the VLMs in a low-data regime. Notably, traditional fine-tuning methods typically involve tuning all parameters of the model to adapt to downstream tasks. However, this approach may not be well-suited for fine-tuning large VLMs in resource-constrained scenarios, and result in overfitting due to the

---

[*]Equal Contribution
[†]Corresponding Authors

37th Conference on Neural Information Processing Systems (NeurIPS 2023).

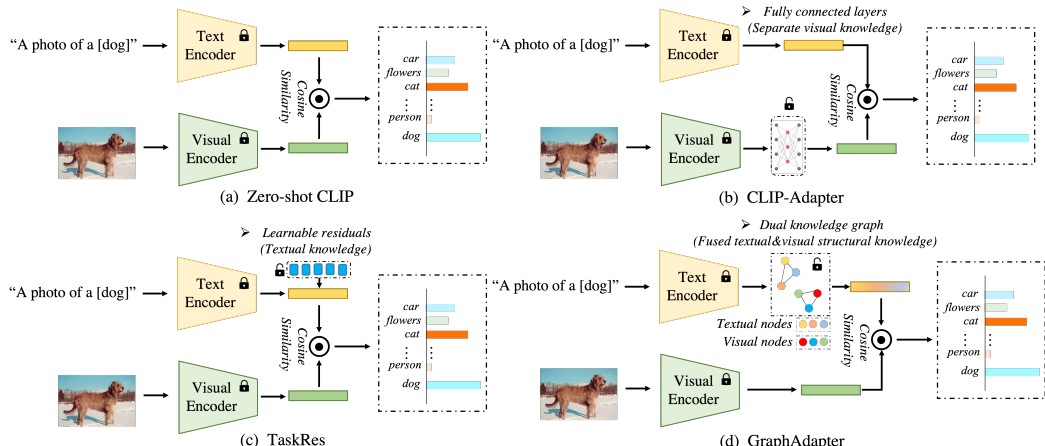

Figure 1: The comparison between (a) Zero-shot CLIP [55], (b) CLIP-Adapter [18] (c) TaskRes [82], and (d) our proposed GraphAdapter. We can observe that previous works model task-specific knowledge with a single modality and lacks the exploitation of structure knowledge. In contrast, our GraphAdapter aims to exploit the fused vision and language structure knowledge in data (*i.e.*, the inter-class relationships in dual modalities) for textual feature adapter with graph learning.

limited number of available samples. To mitigate this, efficient transfer learning (ETL) is proposed to transfer the task-relevant knowledge from VLMs to downstream tasks by tuning a few parameters.

There are two popular ETL approaches for VLMs, including prompt tuning [89, 88, 44, 10, 62, 25, 42, 6], and adapter-style tuning [18, 52, 65, 90, 86, 85]. In particular, prompt tuning aims to adjust the textual classifier adaptively toward downstream tasks by adding learnable prompts on the input side, which outperforms zero-shot CLIP by a large margin with few-shot samples, such as CoOp [89], and CoCoOp [88]. Despite that, there is one limitation in the prompt tuning of VLMs [83, 82]: it needs to pass the data through the textual encoder every iteration during training, yielding a higher demand for resources. In contrast, by using the textual encoder once only, adapter-style works tend to refine the textual classifier or visual features with simple but efficient feature modulation for a specific task on the output side. For instance, CLIP-Adapter [18] exploits one simple bottleneck layer to adjust the textual and visual embeddings of VLMs, which exceeds the zero-shot CLIP by 3.02% on ImageNet with the one-shot setting. TaskRes [82] utilizes learnable task-specific parameters as prior-independent residuals to adjust the textual embeddings. Another popular line [86, 85, 90] seeks to augment the prior knowledge for downstream tasks with the cooperation of CLIP and other pre-trained large vision or language models, such as DINO [5], and GPT [4].

However, there are two limitations in most adapter-style works on ETL: 1) only modeling task-specific knowledge from a single modality perspective, such as CLIP-adapter [18], TaskRes [82] and Tip-Adapter [86], where the adaptation is achieved based on the independent visual or textual feature. 2) overlooking the explicit exploitation of the structure knowledge (*i.e.*, the relation between different semantics/classes) in downstream tasks. A small number of samples in a low-data regime are hard to guide the model to sufficiently excavate the structure knowledge in downstream tasks, leading to the bias for partial attributes in the data, such as color and shape, and causing the sub-optimal transferability and generalization capability [33, 49, 9, 45, 46]. It is vital to model the multi-modality structure knowledge of downstream tasks for the tuning of VLMs in the low-data regime.

To mitigate the above limitations, we propose a brand-new adapter-style tuning strategy, dubbed GraphAdapter, which aims to model task-specific knowledge for downstream tasks with the fused textual and visual structure knowledge. We achieve this by solving two crucial challenges: 1) how to model the structure knowledge (*i.e.*, the inter-class relationship) for downstream tasks, and 2) how to learn the task-specific knowledge by introducing two-modality (*i.e.*, visual and textual) structure knowledge. Recently, graph learning [49, 7, 36] has shown the prevailing performance on modeling data knowledge structure. However, the potential of graphs in improving the efficient transfer learning of VLMs has not yet been fully explored. Inspired by this, for the first challenge, we aim to exploit graph learning to model the structure knowledge for downstream tasks when tuning VLMs. Consequently, we propose the dual knowledge graph, which is composed of a textual sub-graph and a visual sub-graph. Particularly, to establish the relationship between different

semantics/classes in textual space, we regard the textual feature of a specific class as one node of the textual graph and measure inter-class relationships by measuring their corresponding distance of features as edges of the graph. Similarly, the visual knowledge graph aims to model the relationship between different semantics/classes from the visual perspective, where the nodes are constructed with mean features of the training samples from the same class. In this way, our dual knowledge graph contains two-modality structure knowledge from downstream tasks, which enables the feature adapter with sufficient and reliable perception for downstream tasks.

For the second challenge, one intuitive strategy is to introduce the textual/visual structure knowledge for the textual/visual feature adapter, separately. However, this strategy still models the task-specific knowledge with a single modality for each textual/visual adapter (*e.g.*, the textual adapter is only aware of textual structure knowledge). To mitigate this limitation, we introduce the dual knowledge graph for textual/visual adapters, and let each adapter be aware of the structure knowledge in the same modality and cross-modality. (*Notably, we only exploit the textual adapter in our paper, since the limited gain with the combination of textual and visual adapters. Please see Table 2.*). To exploit the dual knowledge graph effectively, we introduce graph learning to the textual adapter, where each feature of the prompt warps the textual and visual structure knowledge through the graph convolution network (GCN), and then fuse two warped knowledge to modulate the original feature for classification in a residual way. Based on the above strategies, we propose an effective adapter-style tuning strategy, *i.e.*, GraphAdapter, which achieves superior performance in tuning the vision-language models under a low-data regime.

The contributions of this paper are summarized as follows:

- We pinpoint that the dual-modality structure knowledge (*i.e.*, the inter-class relationship in textual and visual space) is vital for the efficient transfer learning (ETL) of VLMs in the low-data regime.

- Based on the above analysis, we propose a brand-new adapter-style tuning strategy, *i.e.*, GraphAdapter, for the ETL of VLMs, which models the dual-modality structure knowledge from the textual and visual space with our proposed dual knowledge graph and enables the feature adapter to leverage the fused visual and language knowledge for better learning of task-specific knowledge from downstream tasks, yielding an effective tuning of VLMs.

- We evaluate our Graphadapter on 11 popular benchmarks on few-shot classification. The experiments demonstrated that our Graphadapter significantly outperforms previous prompt-based or adapter-style works, even on the challenging fine-grained image classification task, such as FGVCAircraft [48].

## 2   Related Work

**Vision-Language Pre-training (VLP)** aims to learn the universal textual and visual representation simultaneously with amounts of text-image pairs. A series of works have revealed that pre-trained vision-language representations can help lots of downstream tasks to improve their performances, such as few-shot classification [89, 18, 82], cross-modality generation [50, 56, 54], and visual recognition [69]. The methods of VLP can be roughly divided into three types: 1) dual-encoder architecture, which aligns the textual and visual representation with text-image matching or text-image contrastive learning. One typical work is CLIP [55], where the promising visual representation is learned by matching the visual concepts with the corresponding textual representation with contrastive learning. 2) The second type [66, 43, 64, 34, 84, 72] is fusion-encoder architecture, where the textual and visual representation are fused with cross-modal attention. In general, two independent feature extractors are exploited to extract the modal-specific representations, respectively, and the cross-modal transformer is used for learning the text-image shared representation. 3) The third type is a combination of dual-branches and fusion-branch architectures. For instance, VLMo [1] proposes the multiway transformer, which consists of the visual-specific branch, text-specific branch, and text-visual branch. Recently, amounts of studies are devoted to exploring how to tune the large vision-language model to downstream tasks with few parameters and data, *i.e.*, efficient transfer learning. This will be clarified in the next section.

**Efficient Transfer Learning (ETL)** is proposed to transfer the task-specific knowledge to downstream tasks by tuning the partial parameters [24, 44, 89, 18, 37, 80, 79, 39, 40, 16, 26, 76, 70]. There are two prominent directions for the ETL, *i.e.*, prompt tuning [89, 88, 44] and adapter-style tuning [18, 82, 85, 86]. Prompt engineering stems from natural language process, (NLP), which aims

to formalize various NLP tasks with different prompt templates [38, 47]. Based on this development, prompt engineering is introduced to the visual-language task. For instance, Zero-shot CLIP [55] introduces the simple manual prompt template, like "a photo of a [class]", and outperforms the linear probe by 1.9% on ImageNet [12]. As the pioneering work, CoOp [89] for the first time introduces the learnable prompt to transfer the task-specific knowledge to few-shot tasks. Following this, amounts of work [88, 87, 85, 44, 28, 57, 62, 10, 25, 42, 73, 6] improve the prompt tuning from multiple perspectives, such as generalization [88], knowledge prototype [87], augmentation [85], and diversity [44]. In contrast, adapter-style works tune the VLMs to downstream tasks by adjusting the textual and visual features. Early work CLIP-Adapter [18] only exploits one bottleneck layer to improve the performance of few-shot classification. TaskRes [82] introduces the task-independent adapter to decouple the prior knowledge of the pre-trained models and task-specific knowledge. There are also some works [86, 85] that seek extra prior knowledge from other pre-trained large models, such as DINO [5], which also yield great performance on adapter-style works. In principle, the above adapter-style methods achieve efficient transfer learning for VLMs from two perspectives: 1) increasing instance-wise or task-wise adaptability based on downstream tasks [18, 82]. 2) taking into account the relation between training and test samples. [86]. However, the above works only model task-specific knowledge with a single modality, and lacks the utilization of structure knowledge in downstream tasks, leading to the sub-optimal solution. In this paper, we aim to excavate the structure knowledge of data with graphs and further improve the performance of adapter-style tuning.

**Graph learning** [77, 74, 2, 29, 33, 81] performs excellent capability in formalizing the topological structure of data with the nodes and edges, which has been broadly applied in various fields, such as biology [35], and social networks [59]. With the emergence of GCN [29], some works take a step forward to introduce structure graphs into transfer learning [9, 8, 7, 33, 15]. The core challenge for graph learning is how to construct the nodes and edges. Previous works usually construct the nodes from three perspectives, including objects [9], semantic features [8], or model parameters [7]. There are also some works that utilize the hypergraph to achieve better learning of structure knowledge [15, 78]. In contrast, in this paper, we are the first work to introduce graph learning into the efficient transfer learning (ETL) of vision-language tasks.

## 3 Approach

### 3.1 Preliminaries

**Contrastive Language-Image Pre-training** (CLIP) is designed to learn continuous visual representation based on 0.4 million text-image pairs. Different from the pretraining with ImageNet, where the labels are discrete and cannot delicately describe the attributes of images and the relation between different classes, CLIP learns the visual representation with the supervision of textual embedding by contrastive loss. There are two feature extractors in CLIP, *i.e.*, the textual encoder $E_t$ and visual encoder $E_v$, which are used to warp the prompts and image into the textual embedding $z_t$ and visual embedding $z_v$. In the test process, $K$ prompts like "a photo of a [class]" are inputted to the textual feature extractor to obtain the textual embeddings $\{z_t^k\}_{k=1}^K$ as the labels of $K$ classes. The class of each image is achieved with:

$$P(y = c|z_v) = \frac{exp(sim(z_v, z_t^c)/\tau)}{\sum_{k=1}^K exp(sim(z_v, z_t^k)/\tau)},\tag{1}$$

where $c$, $sim$, and $\tau$ denote the class, cosine similarity, and learned temperature of CLIP, respectively. Despite CLIP has performed excellently on zero-shot image classification, the efficient transfer learning of CLIP is still required to adapt the pre-trained CLIP to downstream tasks.

### 3.2 Our Approach: GraphAdapter

To introduce the dual-modality structure knowledge of the downstream tasks to the feature adapter, we introduce dual knowledge graph $\mathcal{G} = \{\mathcal{G}_t, \mathcal{G}_v\}$, which is composed of the textual knowledge sub-graph $\mathcal{G}_t$ and visual knowledge sub-graph $\mathcal{G}_v$ to obtain and store the textual structure knowledge of prompts and visual structure knowledge of training samples, respectively. Then the feature of the sample/prompt can warp the structure knowledge of downstream tasks in the two modalities to refine themselves, thereby leveraging the dual structure knowledge for better modeling for downstream tasks. The whole framework of our GraphAdapter is depicted in Fig. 3, which is composed of two parts, including the construction of a dual knowledge graph and the adapter-style CLIP-based classification. Particularly, the dual knowledge graph is established only once for the whole training

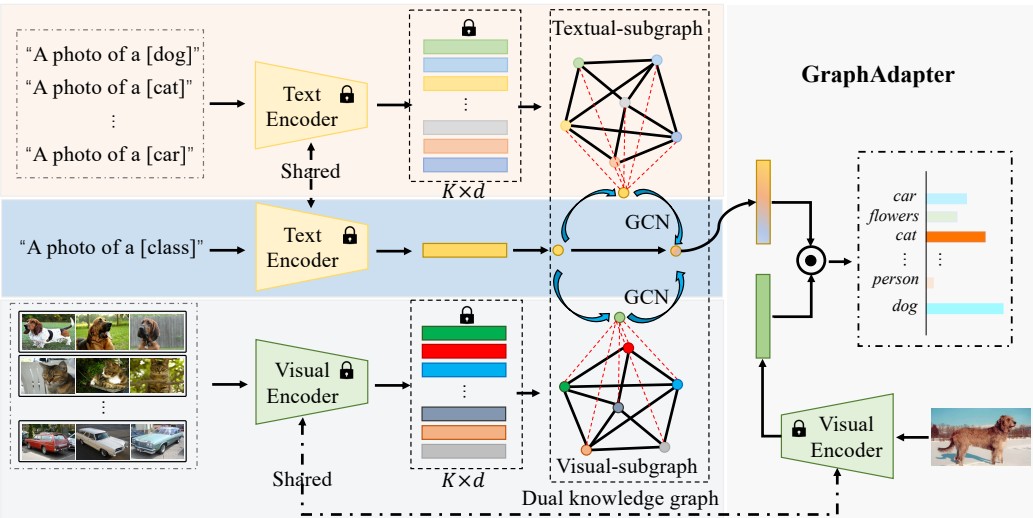

Figure 2: The pipeline of GraphAdapter, which is composed of the dual knowledge graph and CLIP-based image classification pipeline. Here, the dual knowledge graph contains the delicately designed sub-graphs for textual and visual knowledge, respectively, where the textual knowledge sub-graph is established with the textual feature from the prompts of the downstream task, and the visual knowledge sub-graph is constructed with the visual features of training samples from the downstream task. In the optimization process, the feature of each prompt used for classification will seek the fused visual and textual structure knowledge from the dual knowledge graph to adjust themselves for better classification. $K$ and $d$ are the number of classes and the dimension of textual/visual features.

process, and the textual encoder is also only run once to obtain the textual embedding $z_t$ of each prompt. In the optimization process, given one textual feature $z_t$, the relation between it and the nodes of the dual knowledge graph will be computed as the condition to warp the knowledge from the dual knowledge graph to adjust the textual embedding in the residual form. We will clarify our GraphAdapter as follows:

**Textual knowledge sub-graph.** To excavate the structure knowledge (*i.e.*, the relation between different semantics) of the downstream tasks in textual space, we construct the textual knowledge sub-graph $\mathcal{G}_t = \{\mathcal{C}_t, \mathcal{E}_t\}$, where the nodes $\mathcal{C}_t$ aims to capture the semantics of different classes and the edges $\{\mathcal{E}_t\}$ is used to measure the relation of different nodes. Notably, the classifier of CLIP is based on the textual features of the prompts from different classes, where the textual feature of a within-class prompt can represent the semantics of one class. Therefore, given one downstream task with $K$ classes, the nodes set $\mathcal{C}_t = \{c_t^i\}_{i=1}^K \in \mathbb{R}^{K \times d})$ are obtained with the mean feature of the prompts from the different class, where $c_t^i$ and $d$ denotes the textual feature of the $i^{th}$ class and the dimension of the textual feature. And the edges $\mathcal{E}_t$ between different nodes are computed with the cosine similarity between different nodes since the classification is achieved by computing the cosine similarity between the features in CLIP.

$$\mathcal{E}_t = \{e_t^{i,j}\}, e_t^{i,j} = \frac{c_t^i c_t^{j^T}}{|c_t^i| \cdot |c_t^j|}, i, j \in [1, K], \tag{2}$$

where $\{e_t^{i,j}\}$ denotes the edge between $i^{th}$ and $j^{th}$ nodes.

**Visual knowledge sub-graph.** Different from the textual knowledge sub-graph, where the structure knowledge is only from the textual concept of a downstream task, the visual knowledge sub-graph can measure more fine-grained relations of different semantics in visual space, since the diverse visual feature of different samples. As shown in Fig. 3, to construct the visual knowledge sub-graph $\mathcal{G}_v = \{\mathcal{C}_v, \mathcal{E}_v\}$, we pass the augmented image group from the same class into visual encoder to obtain their visual features, and then compute the mean features of them as the nodes $\mathcal{C}_v = \{c_v^i\}_{i=1}^K \in \mathbb{R}^{K \times d}$ of for visual knowledge graph. The edges $\mathcal{E}_v = \{e_v^{i,j} | i, j \in [1, K]\}$ are computed with the cosine similarity between different nodes in the visual knowledge sub-graph.

**Adapting with the dual knowledge graph.** After constructing the dual knowledge graph $\mathcal{G} = \{\mathcal{G}_t, \mathcal{G}_v\}$, we can achieve the feature adapter by introducing the dual-modality structure knowledge adaptively from the text and image sub-graphs. Notably, previous works on adapter-style tuning only model the task-specific knowledge with a single modality, which lacks the exploitation of cross-modality knowledge. In contrast, our GraphAdapter utilizes both inner-modality and cross-modality structure knowledge for feature adapters. Concretely, given the textual feature $z_t$ from the textual encoder of CLIP, we aim to warp the $z_t$ into the textual and visual knowledge sub-graphs to extract the textual modality and cross-modality structure knowledge, respectively. One simple strategy is to regard the textual/visual features $z_t$ and $z_v$ as the query, and then warp the knowledge from the graph nodes $\mathcal{C}_t$ and $\mathcal{C}_v$ based on similarity. However, this ignores that the structure knowledge is also required to be optimized to suit the downstream tasks. To achieve this, in the adapting process, we regard the textual feature $z_t$ as one node and then warp it to the visual and textual sub-graphs, where the textual features obtain dual-modality structure knowledge by interacting with two sub-graphs in the same graph space with the graph convolutional networks (GCN).

Specifically, in the optimization process, we expand the textual knowledge sub-graph and visual knowledge sub-graph by concatenating the nodes $\mathcal{C}_t/\mathcal{C}_v$ of the textual/visual knowledge sub-graph with the textual features $z_t$, and then compute their edges (*i.e.*, the correlation between different nodes) as:

$$\mathcal{C}_{tt} = [z_t, \mathcal{C}_t], \quad \mathcal{C}_{vt} = [z_t, \mathcal{C}_v],$$
$$\mathcal{E}_{tt} = \begin{bmatrix} 1 & \text{sim}(z_t, \mathcal{C}_t) \\ \text{sim}(\mathcal{C}_t, z_t) & \mathcal{E}_t \end{bmatrix}, \quad \mathcal{E}_{vt} = \begin{bmatrix} 1 & \text{sim}(z_t, \mathcal{C}_v) \\ \text{sim}(\mathcal{C}_v, z_t) & \mathcal{E}_v \end{bmatrix}. \tag{3}$$

In this way, we can warp the textual feature to the dual knowledge graph space, which provides the basis for interaction and knowledge transfer between the textual feature and the dual knowledge graph. Then, the Graph Convolution Network (GCN) $g_{tt}$ and $g_{vt}$ are utilized to excavate the knowledge for adapting the texture feature from each node in textual and visual knowledge sub-graphs:

$$\mathcal{C}_{tt}^* = g_{tt}(\mathcal{C}_{tt}, \hat{\mathcal{E}}_{tt}) = \sigma(\hat{\mathcal{E}}_{tt}\mathcal{C}_{tt}W_{tt}), \quad \mathcal{C}_{vt}^* = g_{vt}(\mathcal{C}_{tt}, \hat{\mathcal{E}}_{vt}) = \sigma(\hat{\mathcal{E}}_{vt}\mathcal{C}_{vt}W_{vt}), \tag{4}$$

where $\hat{\mathcal{E}}_{tt} = D_{tt}\mathcal{E}_{tt}D_{tt}$ and $\hat{\mathcal{E}}_{vt} = D_{vt}\mathcal{E}_{vt}D_{vt}$ are the adjacent matrix for graph learning. $\sigma$ is an activation function. And the $D_{tt}$ and $D_{vt}$ are the matrices used for Laplace normalization [29] for the edges as:

$$D_{tt} = \text{diag}(\sum_{p=1}^{K}(\mathcal{E}_{tt} + I)_p), \quad D_{vt} = \text{diag}(\sum_{p=1}^{K}(\mathcal{E}_{vt} + I)_p). \tag{5}$$

The meaning of Eq. 4 is seeking the prior knowledge from other nodes in one graph to refine itself based on the correlation (*i.e.*, the adjacent matrix). And thus, we can obtain the adapted/refined textual feature as $z_{tt} = \mathcal{C}_{tt}^*[0,:]$ and $z_{vt} = \mathcal{C}_{vt}^*[0,:]$. To fuse the inner-modality and cross-modality structure knowledge, we introduce one hyper-parameter $\beta$ to fuse these two features as $z_t' = \beta * z_{tt} + (1-\beta) * z_{vt}$. Then the feature adapter is achieved in a residual form, where the hyper-parameter $\alpha$ is used to adjust the weights of prior knowledge from the dual knowledge graph and original feature: $z_t^* = \alpha z_t + (1-\alpha)z_t'$.

In the whole training process, only the parameters of GCN $g_{tt}$ and $g_{vt}$ are trainable under the constraint of the cross entropy function between the label of samples and the predicted label as Eq. 1.

## 4 Experiments

### 4.1 Datasets and Implementation Details

**Datasets.** Following previous adapter-style studies [18, 82, 86], we validate our GraphAdapter on 11 few-shot classification tasks, including ImageNet [12], StandfordCars [30], UCF101 [63], Caltech101 [17], Flowers102 [51], SUN397 [75], DTD [11], EuroSAT [21], FGVCAircraft [48], OxfordPets [53], and Food101 [3]. Among them, OxfordPets, Food101, StanfordCars, Flowers102, and FGVCAircraft belong to fine-grained classification tasks, EuroSAT is for remote sensing classification, and DTD is the dataset of texture classification. To investigate the generalization capability of our GraphAdapter, we follow the CoOp [89] and conduct experiments on ImageNetV2 [58], ImageNet-Sketch [68], ImageNet-A [23] and ImageNet-R [22].

**Implementation details.** If not mentioned, we set the pre-trained backbone ResNet-50 of CLIP [19] as the base layer to produce the visual feature. To estimate the applicability of our method, we also

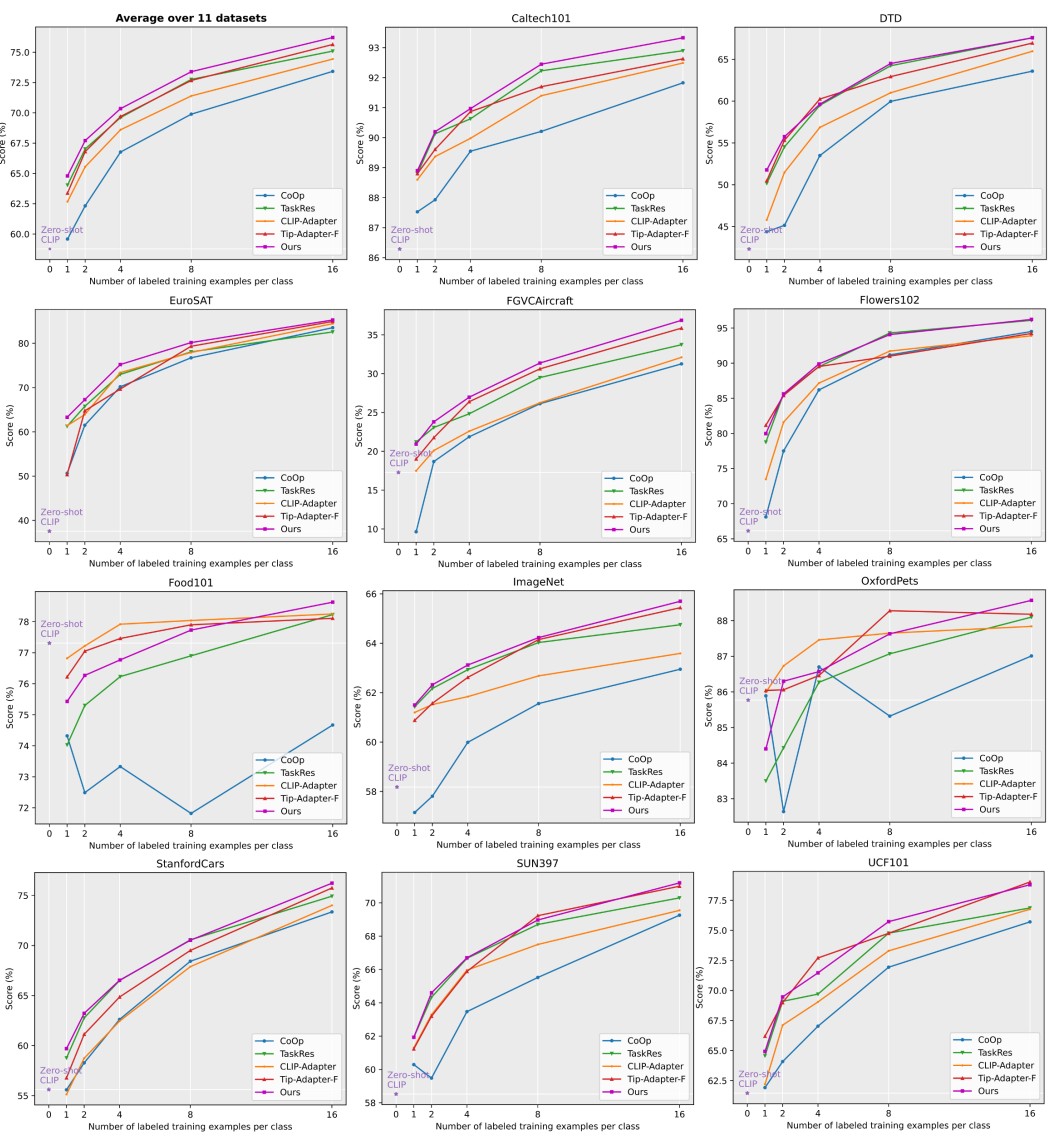

Figure 3: The performance comparison of our GraphAdapter with the state-of-the-art methods on few-shot learning, including 1-/2-/4-/8-/16-shots on 11 benchmark datasets. We provide all numerical results of this figure in the Supplementary.

apply our GraphAdapter to other CLIP visual encoders, including ResNet-101 [20], ViT-B/32 [14], and ViT-B/16 [14], and validate its effectiveness on ImageNet. We optimize our model for 100 epochs for 1, 2, 4, 8, and 16-shots. In the training process, we utilize the Adam optimizer with an initial learning rate of $1e^{-3}$, which drops with the cosine learning rate decay schedule. Notably, to achieve stable training, we follow previous works [82, 89] and utilize the warmup strategy for the training, where the small learning rate $1e^{-5}$ is applied at the first epoch. The visual sub-graph is constructed with the few-shot training samples. Before the training, we exploit the visual encoder of pre-trained CLIP to extract the visual features for these few-shot samples from the same class and average them as the node of this class. The number of images used for each class is decided by the number of shots. For example, for the 2-shot task, two images are used for each node. The data augmentation strategy only contains "random resized cropping" and "random flipping".

## 4.2 Comparisons with State-of-the-arts

**Few-shot learning.** We compare our proposed GraphAdapter with several state-of-the-art works on ETL, including Zero-shot CLIP [55], CoOp [89], clip-adapter [18], Tip-Adapter-F [86], and

Table 1: The performance comparison in terms of generalization capability on four CLIP visual backbones. The ETL methods are optimized with the ImageNet dataset on 16-shot setting and tested on cross-domain datasets, including ImageNet-V2, -Sketch, -A, and -R.

| Method | Backbone | Source | Target | | | | |
|---|---|---|---|---|---|---|---|
| | | ImageNet | -V2 | -Sketch | -A | -R | Average |
| Zero-shot CLIP [55] | | 58.18 | 51.34 | 33.32 | 21.65 | 56.00 | 40.58 |
| Linear Probe CLIP [55] | | 55.87 | 45.97 | 19.07 | 12.74 | 28.16 | 28.16 |
| CoOp [89] | | 62.95 | 55.11 | 32.74 | 22.12 | 54.96 | 41.23 |
| TaskRes [82] | ResNet-50 | 64.75 | 56.47 | 35.83 | 22.80 | 60.70 | 43.95 |
| Ours | | **65.70** | 56.40 | 34.50 | 21.88 | 58.94 | 42.93 |
| Ours$_g$ | | 64.94 | **56.58** | **35.89** | **23.07** | **60.86** | **44.10** |
| Zero-shot CLIP [55] | | 61.62 | 54.81 | 38.71 | 28.05 | 64.38 | 46.49 |
| Linear Probe CLIP [55] | | 59.75 | 50.05 | 26.80 | 19.44 | 47.19 | 35.87 |
| CoOp [89] | | 66.60 | 58.66 | 39.08 | 28.89 | 63.00 | 47.41 |
| TaskRes [82] | ResNet-101 | 67.70 | 59.50 | **41.70** | 29.87 | 68.07 | 49.79 |
| Ours | | **68.23** | **59.60** | 40.83 | 28.77 | 67.13 | 49.08 |
| Ours$_g$ | | 67.87 | 59.50 | 41.60 | **30.00** | **68.10** | **49.80** |
| Zero-shot CLIP [55] | | 62.05 | 54.79 | 40.82 | 29.57 | 65.99 | 47.79 |
| Linear Probe CLIP [55] | | 59.58 | 49.73 | 28.06 | 19.67 | 47.20 | 36.17 |
| CoOp [89] | | 66.85 | 58.08 | 40.44 | 30.62 | 64.45 | 48.40 |
| TaskRes [82] | ViT-B/32 | 68.20 | **59.20** | 42.50 | 31.43 | 69.33 | 50.62 |
| Ours | | **68.80** | 59.00 | 41.70 | 29.57 | 68.67 | 49.74 |
| Ours$_g$ | | 68.47 | 59.10 | **42.70** | **31.73** | **69.43** | **50.74** |
| Zero-shot CLIP [55] | | 66.73 | 60.83 | 46.15 | 47.77 | 73.96 | 57.18 |
| Linear Probe CLIP [55] | | 65.85 | 56.26 | 34.77 | 35.68 | 58.43 | 46.29 |
| CoOp [89] | | 71.92 | 64.18 | 46.71 | 48.41 | 74.32 | 58.41 |
| TaskRes [82] | ViT-B/16 | 73.07 | 65.30 | 49.13 | 50.37 | 77.70 | 60.63 |
| Ours | | **73.68** | 65.57 | 48.57 | 49.23 | 77.20 | 60.14 |
| Ours$_g$ | | 73.40 | **65.60** | 49.23 | 50.57 | 77.73 | 60.78 |

TaskRes [82] on 11 benchmark datasets. The experimental results are shown in Fig. 3, where we can observe that our GraphAdapter consistently outperforms previous ETL works for 1-/2-/4-/8-/16-shots on the average performance of 11 benchmark datasets. Particularly, on the 16-shot setting, our GraphAdapter achieves an average performance of 76.22%, which exceeds the Tip-Adapter-F [86] by 0.57%, and TaskRes [82] by 1.12%. Even for the most challenging dataset FGVCAircraft [48] of the fine-grained classification, our GraphAdapter still performs better than the second-best method Tip-Adapter-F by 1.01% on 16-shot setting, since our GraphAdapter can better exploit the structure knowledge in the downstream tasks with the dual knowledge graph.

**Generalization.** We also investigate the generalization capability of our GraphAdapter on four commonly-used datasets *i.e.*, ImageNet-V2 [58], ImageNet-Sketch [68], ImageNet-A [23], ImageNet-R [22], with different visual backbones, including the pre-trained ResNet-101 [20], ViT-B/16 [13], ViT-B/32 [13]. The experimental results are shown in Table 1, where we provide two versions "Ours$_g$" and "Ours" of our GraphAdapter for a fair comparison. Here, "Ours" denotes the version of GraphAdapter used in few-shot learning. Since the over-fitting on few-shot learning will decrease the generalization capability, we increase the weights $\alpha = 0.8$ of base textual features $z_t$ in the adapted feature $z_t^*$ as "Ours$_g$" to obviate the over-fitting on ImageNet Dataset. From Table 1, our GraphAdapter achieves the optimal generalization capability on four cross-domain datasets, which outperforms TaskRes [82] by 0.15% on ResNet-50 [20]. Meanwhile, it obtains the best performance on the source data ImageNet.

## 4.3 Ablation Studies

**The effects of the different variants of GraphAdapter.** It is noteworthy that our GraphAdapter can be applied to the textual and visual adapter by introducing fused visual and textual structure knowledge. Therefore, we investigate the effectiveness of another two variants of our GraphAdapter, *i.e.*, GraphAapter-I, and GraphAdapter-T&I. Here, we denote our GraphAdapter in our paper as GraphAdapter-T, since it is used to achieve the feature adapter of the textual branch. GraphAapter-I denotes that we introduce the GraphAdapter into the visual branch, which adjusts the visual feature to adapt the downstream tasks. GraphAdapter-T&I represents that the GraphAdapter is exploited in the visual and textual branches simultaneously. As shown in Table 2, we can have the following conclusions: 1) the GraphAdapter is more suitable for the textual branch, which outperforms that in

Table 2: The ablation study for different variants of our GraphAdapter, where GraphAdapter-T, GraphAdapter-I denotes integrating our proposed GraphAdapter into the textual branch and visual branch, respectively. GraphAdapter-T&I utilizes the GraphAdapter-T and GraphAdapter-I simultaneously.

| Method \ Dataset | ImageNet | Caltech101 | OxfordPets | StanfordCars | Flowers102 | Food101 | FGVCAircraft | SUN397 | DTD | EuroSAT | UCF101 | Average |
|---|---|---|---|---|---|---|---|---|---|---|---|---|
| CLIP-Adapter [18] | 63.59 | 92.49 | 87.84 | 74.01 | 93.90 | 78.25 | 32.10 | 69.55 | 65.96 | 84.43 | 76.76 | 74.44 |
| CoOp [89] | 62.95 | 91.83 | 87.01 | 73.36 | 94.51 | 74.67 | 31.26 | 69.26 | 63.58 | 83.53 | 75.71 | 73.42 |
| TaskRes [82] | 64.75 | 92.90 | 88.10 | 74.93 | 96.10 | 78.23 | 33.73 | 70.30 | 67.57 | 82.57 | 76.87 | 75.10 |
| Tip-Adapter [86] | 65.44 | 92.63 | 88.18 | 75.75 | 94.23 | 78.11 | 35.86 | 71.00 | 66.94 | 84.94 | 79.03 | 75.65 |
| Base | 58.18 | 86.29 | 85.77 | 55.61 | 66.14 | 77.31 | 17.28 | 58.52 | 42.32 | 37.56 | 61.46 | 58.77 |
| GraphAdapter-T | 65.70 | 93.33 | 88.57 | **76.23** | **96.23** | 78.63 | 36.87 | 71.20 | 67.57 | 85.27 | **78.80** | 76.22 |
| GraphAdapter-I | 63.50 | 92.10 | 85.86 | 71.81 | 93.24 | 74.55 | 30.90 | 67.39 | 61.75 | 78.79 | 74.40 | 72.21 |
| GraphAdapter-T&I | **65.72** | **93.37** | **88.59** | 76.20 | 96.13 | 78.40 | **36.93** | **71.30** | **67.90** | **85.55** | 78.67 | **76.25** |

Table 3: The ablation studies for two coefficients $\alpha$ and $\beta$.

| $\beta\ (\alpha = 0.6)$ | 0.0 | 0.3 | 0.5 | 0.7 | 1.0 | Learnable |
|---|---|---|---|---|---|---|
| ImageNet | 64.30 | 65.1 | 65.50 | 65.70 | 64.95 | 64.35 |

| $\alpha\ (\beta = 0.7)$ | 0.5 | 0.6 | 0.7 | 0.8 | 0.9 | Learnable |
|---|---|---|---|---|---|---|
| ImageNet | 65.30 | 65.70 | 65.37 | 64.94 | 63.03 | 65.42 |

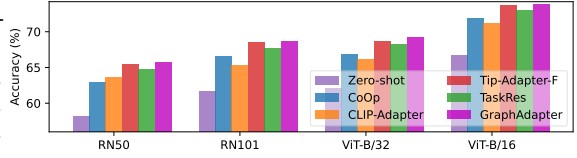

Figure 4: Comparisons on different backbones.

the visual branch by an average of 4.01% on 11 benchmark datasets. 2) Applying the GraphAdapter in both branches of CLIP only achieves a slight gain compared with the GraphAdapter-T. Therefore, we utilize the GraphAdapter-T in our paper.

**The effects of the dual knowledge graph.** To investigate the effectiveness of the dual knowledge graph, we conduct experiments on ImageNet by removing the textual knowledge sub-graph (*i.e.*, setting $\beta$ as 0.0) and visual knowledge sub-graph (*i.e.*, setting $\beta$ as 1.0), respectively. The experimental results are shown in Table 3. We can observe that removing the textual knowledge sub-graph achieves an accuracy of 64.30%, while the performance of removing the visual knowledge sub-graph drops to 64.95%, which reveals that textual structure knowledge is more important for textual feature adapter. Notably, the cooperation of textual and visual structure knowledge can achieve the best performance of 65.70%. This validates the effectiveness of introducing inner-modality and cross-modality structure knowledge for adapter-style tuning.

**The effects of different coefficients $\alpha$ and $\beta$.** There are two hyper-parameters $\alpha$ and $\beta$ in our GraphAdapter, where the $\alpha$ controls the balance of basic textual features from CLIP and the adapted textual feature obtained with GraphAdapter, and $\beta$ is responsible for the weights of textual knowledge graph and visual knowledge graph. As shown in table 3, the performance of GraphAdapter increases first and then drops when the coefficients $\alpha$ and $\beta$ increase. The optimal $\beta$ and $\alpha$ are 0.7, and 0.6, respectively. We also investigate whether a learnable coefficient can further improve the performance of our GraphAdapter. Experimental results show that the learnable coefficients achieve lower performance with our selected fixed coefficients $\alpha$ and $\beta$, since the coefficients are hard to be optimized with the few-shot samples.

**The effects of different backbones.** As stated in Fig. 4, we also evaluate the effectiveness of our GraphAdapter on different CLIP visual backbones, including ResNet-50, ResNet-101, ViT-B/32, and ViT-B/16. Our GraphAdapter consistently exceeds previous works for both four visual backbones.

## 4.4 Complexity Analysis

We compute and compare our methods with existing published efficient transfer learning methods on the ImageNet [12] with the 16-shot setting, from the perspectives of tunable parameters, computational flops, training time, and inference time. All results are measured with the officially released code from GitHub. The experimental results are shown in the Table 4. We can observe our tunable parameters are still less than Tip-Adapter-F [86]. For computational flops, our GraphAdapter takes about 5.42 GFlops, almost the same as Tip-Adapter-F and TaskRes [82], which is far lower than CLIP-Adapter.

Moreover, our training and inference times are less than the adapter-based work CLIP-Adapter and the prompt tuning method CoOp [89]. Therefore, our GraphAdapter satisfies the requirement of efficient transfer learning. Moreover, our GraphAdapter can achieve the best performance.

Table 4: A comparison between our GraphAdapter and existing methods on time and model complexity.

| Methods | CoOp [89] | CLIP-Adapter [18] | Tip-Adapter-F [86] | TaskRes [82] | Ours |
|---|---|---|---|---|---|
| Tunable Parameters (M) | 0.008 | 0.524 | 16.384 | 1.024 | 4.145 |
| GFlops | 1943.12 | 1959.44 | 5.43 | 5.42 | 5.42 |
| Training time (one epoch) (s) | 40.91 | 45.71 | 12.36 | 13.64 | 23.29 |
| Inference time (s/100) | 119.64 | 275.22 | 51.03 | 4.89 | 4.91 |
| Memory Cost (Training) | 18.907 | 9.257 | 4.313 | 6.227 | 10.75 |
| Memory Cost (Inference) | 7.403 | 7.615 | 4.161 | 6.225 | 4.433 |
| Performance | 62.95 | 63.59 | 65.44 | 64.75 | 65.70 |

## 4.5 Applicability

To validate the applicability of our GraphAdapter, we select two state-of-the-art adapter-style works, including CaFo [85] and TaskRes* [82]. Here, CaFo [85] incorporates diverse prior knowledge from large pre-trained vision and language models, including DINO's vision-contrastive knowledge, GPT-3's language-generative knowledge, and DALLE's generative capability. The adapting strategy of CaFo [85] is from the Tip-Adapter [86]. The TaskRes* denotes the enhanced version of TaskRes [82], which exploits the enhanced base classifier instead of the original classifier from CLIP [55].

For CaFo [85], we directly incorporate our GraphAdapter into the textual classifier. For TaskRes* [82], we replace the task residual with our proposed GraphAdapter and maintain its enhanced textual branch from CLIP. The experimental results on ImageNet [12] are shown in Table 5 of Appendix. We can observe that our GraphAdapter can consistently increase the performance of CaFo [85] and TaskRes* [82] on few-shot learning with all 1-/2-/4-/8-/16-shots settings. Particularly, on the 16-shot setting, ours improves CaFo [85] by 0.51%, and TaskRes* by 1.15%, which validates the powerful applicability of our GraphAdapter. *Overall, our GraphAdapter is complementary to these prior-augmented methods, and can obtain better performance by integrating ours into them.*

## 5 Conclusion

In this paper, we comprehensively review the limitations of previous adapter-style tuning methods in the low-data regime as 1) previous works only model the task-specific knowledge with a single modality, and 2) overlooking the exploitation of the structure knowledge (*i.e.*, the relation of different semantics/classes) in downstream tasks, which is vital for data-efficient tasks. Based on the analysis, we propose a brand-new adapter-style tuning strategy for visual-language models, termed as GraphAdapter, which introduces the dual knowledge graph to establish the structure knowledge in the textual and visual modalities. By incorporating the fused textual and visual structure knowledge with Graph Convolutional Network (GCN), the text-based classifier can be adapted to downstream tasks effectively with the inner-modality and cross-modality structure knowledge. Extension experiments on 11 benchmark datasets revealed the effectiveness of our GraphAdapter on few-shot learning and generalization.

**Limitations and Broader Impacts.** The limitation of our GraphAdapter stems from the textual structure knowledge modeling. In this paper, we utilize the textual features of default prompts like "a photo of a [class]" to construct the nodes of the textual graph. However, the prompts are simple and lack enough diversity. We believe more diverse and accurate prompts for downstream tasks can achieve better modeling for textual structure knowledge, which can further improve the performance of our GraphAdapter.

The adapter-style tuning of VLMs aims to efficiently finetune the VLMs for downstream tasks by optimizing a few parameters in the low-data regime. The possible broader impact of our GraphAdapter stems from the tuning of VLMs itself, which has a heavy dependency on the pre-trained VLMs. The utilization of our GraphAdapter should follow the privacy and safety of datasets and pre-trained models.

## Acknowledgement

This research is supported by the National Research Foundation, Singapore, under its AI Singapore Programme (AISG Award No: AISG2-RP-2021-023) and NSFC under Grant U1908209, 62021001.

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

## A  Applicability

Table 5: The experiments for the applicability of our GraphAdapter. For Cafo [85], we incorporate our GraphAdapter into the textual classifier. Notably, the TaskRes* exploits the enhanced base classifier. Therefore, TaskRes* + Ours denotes that TaskRes* replace the task residual with our proposed GraphAdapter.

| Methods | 1-shot | 2-shot | 4-shot | 8-shot | 16-shot |
|---|---|---|---|---|---|
| CaFo [85] | 63.80 | 64.34 | 65.64 | 66.86 | 68.79 |
| +Ours | **63.81** | **64.97** | **66.17** | **67.68** | **69.30** |
| TaskRes* [82] | 61.43 | 62.17 | 62.93 | 64.03 | 64.75 |
| +Ours | **61.73** | **62.53** | **63.47** | **64.57** | **65.80** |

## B  More Experimental Results

We present the numerical results of "Figure 3 in the main text" as Table 6. We compare our GraphAdapter with the state-of-the-art works, including the prompt-based method CoOp [89], and adapter-style methods, *i.e.*, CLIP-Adapter [18], Tip-Adapter-F [86], and TaskRes [82]. Here, the performance of Tip-Adapter-F is reproduced by [82], which aims to ensure a fair comparison with CoOp [89]. From the table, we can find that on the 16-shot few-shot learning, our GraphAdapter outperforms all previous works except for UCF101 [63] where its performance is comparable. Depart from that, for the average accuracy of 11 benchmark datasets in the 1-/2-/4-/8-/16-shot few-shot learning, our GraphAdapter surpasses previous works with a consistent improvement of $0.57\%$ to $0.76\%$. We also make the analysis for the Error Bars by providing the standard deviation (Std) of our experimental results in Table 6.

Table 6: A numerical comparison between our GraphAdapter and the state-of-the-art methods.

| Methods | Setting | Caltech101 | DTD | EuroSAT | FGVCAircraft | Flowers102 | Food101 | ImageNet | OxfordPets | StanfordCars | SUN397 | UCF101 | Avg. |
|---|---|---|---|---|---|---|---|---|---|---|---|---|---|
| Zero-shot CLIP [55] | | 86.29 | 42.32 | 37.56 | 17.28 | 66.14 | 77.31 | 58.18 | 85.77 | 55.61 | 58.52 | 61.46 | 58.77 |
| CoOp [89] | | 87.53 | 44.39 | 50.63 | 9.64 | 68.12 | 74.32 | 57.15 | 85.89 | 55.59 | 60.29 | 61.92 | 59.59 |
| CLIP-Adapter [18] | | 88.60 | 45.80 | 61.40 | 17.49 | 73.49 | **76.82** | 61.20 | 85.99 | 55.13 | 61.30 | 62.20 | 62.67 |
| Tip-Adapter-F [86] | 1-shot | 88.80 | 50.49 | 50.34 | 19.01 | **81.17** | 76.22 | 60.88 | **86.04** | 56.78 | 61.23 | **66.19** | 63.38 |
| TaskRes [82] | | 88.80 | 50.17 | 61.27 | **21.20** | 78.77 | 74.03 | 61.43 | 83.50 | 58.77 | **61.93** | 64.57 | 64.04 |
| Ours (w/ Std) | | **88.90** | **51.77** | **63.30** | 20.93 | 79.98 | 75.43 | **61.50** | 84.40 | **59.70** | **61.93** | 64.93 | **64.80** |
| | | (±0.22) | (±1.48) | (±1.96) | (±0.25) | (±0.90) | (±0.14) | (±0.09) | (±1.02) | (±0.45) | (±0.26) | (±0.59) | (±0.34) |
| Zero-shot CLIP [55] | | 86.29 | 42.32 | 37.56 | 17.28 | 66.14 | **77.31** | 58.18 | 85.77 | 55.61 | 58.52 | 61.46 | 58.77 |
| CoOp [89] | | 87.93 | 45.15 | 61.50 | 18.68 | 77.51 | 72.49 | 57.81 | 82.64 | 58.28 | 59.48 | 64.09 | 62.32 |
| CLIP-Adapter [18] | | 89.37 | 51.48 | 63.90 | 20.10 | 81.61 | 77.22 | 61.52 | **86.73** | 58.74 | 63.29 | 67.12 | 65.55 |
| Tip-Adapter-F [86] | 2-shot | 89.61 | 55.32 | 64.76 | 21.76 | 85.40 | 77.05 | 61.57 | 86.06 | 61.13 | 63.19 | 68.99 | 66.80 |
| TaskRes [82] | | 90.13 | 54.53 | 65.77 | 23.07 | 85.63 | 75.30 | 62.17 | 84.43 | 62.77 | 64.33 | 69.10 | 67.02 |
| Ours (w/ Std) | | **90.20** | **55.75** | **67.27** | **23.80** | 85.63 | 76.27 | **62.32** | 86.30 | **63.23** | **64.60** | **69.47** | **67.71** |
| | | (±0.22) | (±1.56) | (±1.57) | (±0.65) | (±0.25) | (±0.12) | (±0.17) | (±0.99) | (±0.12) | (±0.33) | (±0.42) | (±0.31) |
| Zero-shot CLIP [55] | | 86.29 | 42.32 | 37.56 | 17.28 | 66.14 | 77.31 | 58.18 | 85.77 | 55.61 | 58.52 | 61.46 | 58.77 |
| CoOp [89] | | 89.55 | 53.49 | 70.18 | 21.87 | 86.20 | 73.33 | 59.99 | 86.70 | 62.62 | 63.47 | 67.03 | 66.77 |
| CLIP-Adapter [18] | | 89.98 | 56.86 | 73.38 | 22.59 | 87.17 | **77.92** | 61.84 | **87.46** | 62.45 | 65.96 | 69.05 | 68.61 |
| Tip-Adapter-F [86] | 4-shot | 90.87 | **60.25** | 69.66 | 26.39 | 89.53 | 77.46 | 62.62 | 86.86 | 64.86 | 65.88 | **72.71** | 69.70 |
| TaskRes [82] | | 90.63 | 59.50 | 72.97 | 24.83 | 89.50 | 76.23 | 62.93 | 86.27 | 66.50 | 66.67 | 69.70 | 69.61 |
| Ours (w/ Std) | | **90.97** | 59.63 | **75.20** | **26.97** | **89.90** | 76.77 | **63.12** | 86.57 | **66.53** | **66.70** | 71.47 | **70.35** |
| | | (±0.05) | (±0.39) | (±1.37) | (±0.29) | (±0.19) | (±0.26) | (±0.19) | (±1.47) | (±0.29) | (±0.28) | (±0.16) | (±0.27) |
| Zero-shot CLIP [55] | | 86.29 | 42.32 | 37.56 | 17.28 | 66.14 | 77.31 | 58.18 | 85.77 | 55.61 | 58.52 | 61.46 | 58.77 |
| CoOp [89] | | 90.21 | 59.97 | 76.73 | 26.13 | 91.18 | 71.82 | 61.56 | 85.32 | 68.43 | 65.52 | 71.94 | 69.89 |
| CLIP-Adapter [18] | | 91.40 | 61.00 | 77.93 | 26.25 | 91.72 | **78.04** | 62.68 | 87.65 | 67.89 | 67.50 | 73.30 | 71.40 |
| Tip-Adapter-F [86] | 8-shot | 91.70 | 62.93 | 79.33 | 30.62 | 91.00 | 77.90 | 64.15 | **88.28** | 69.51 | **69.23** | 74.76 | 72.67 |
| TaskRes [82] | | 92.23 | 64.23 | 78.07 | 29.50 | **94.30** | 76.90 | 64.03 | 87.07 | **70.57** | 68.70 | 74.77 | 72.76 |
| Ours (w/ Std) | | **92.45** | **64.50** | **80.17** | **31.37** | 94.07 | 77.73 | **64.23** | 87.63 | 70.53 | 68.97 | **75.73** | **73.40** |
| | | (±0.38) | (±0.34) | (±1.87) | (±0.40) | (±0.12) | (±0.19) | (±0.08) | (±0.26) | (±0.12) | (±0.12) | (±0.45) | (±0.29) |
| Zero-shot CLIP [55] | | 86.29 | 42.32 | 37.56 | 17.28 | 66.14 | 77.31 | 58.18 | 85.77 | 55.61 | 58.52 | 61.46 | 58.77 |
| CoOp [89] | | 91.83 | 63.58 | 83.53 | 31.26 | 94.51 | 74.67 | 62.95 | 87.01 | 73.36 | 69.26 | 75.71 | 73.42 |
| CLIP-Adapter [18] | | 92.49 | 65.96 | 84.43 | 32.10 | 93.90 | 78.25 | 63.59 | 87.84 | 74.01 | 69.55 | 76.76 | 74.44 |
| Tip-Adapter-F [86] | 16-shot | 92.63 | 66.94 | 84.94 | 35.86 | 94.23 | 78.11 | 65.44 | 88.18 | 75.75 | 71.00 | **79.03** | 75.65 |
| TaskRes [82] | | 92.90 | **67.57** | 82.57 | 33.73 | 96.10 | 78.23 | 64.75 | 88.10 | 74.93 | 70.30 | 76.87 | 75.10 |
| Ours (w/ Std) | | **93.33** | **67.57** | **85.27** | **36.87** | **96.23** | **78.63** | **65.70** | **88.57** | **76.23** | **71.20** | 78.80 | **76.22** |
| | | (±0.08) | (±0.09) | (±0.29) | (±0.50) | (±0.16) | (±0.08) | (±0.08) | (±0.51) | (±0.17) | (±0.08) | (±0.26) | (±0.11) |

## C More Dataset and Implementation Details

**More Dataset Details.** In this paper, we follow previous works, *e.g.*, CoOp [89], CLIP-Adapter [18], TaskRes [82], and Tip-Adapter [86], and exploit the prompts in Table 7 for the tuning and testing.

**More Implementation Details.** Our experimental results are achieved by running the algorithm three times with different seeds for each setting. The training and inference are implemented with a single NVIDIA GeForce RTX 3090. In the implementation of GraphAdapter for the ImageNet [12], we decouple the sub-graph with 1000 nodes for each modality into four graphs with 256 nodes to alleviate the computational cost.

Table 7: The number of classes and the used prompt temple for each dataset.

| Datasets | # Classes | Prompt Templet |
|---|---|---|
| Caltech101 [17] | 100 | "a photo of a [class]." |
| DTD [11] | 47 | "[class] texture." |
| EuroSAT [21] | 10 | "a centered satellite photo of [class]." |
| FGVCAircraft [48] | 100 | "a photo of a [class], a type of aircraft." |
| Flowers102 [51] | 102 | "a photo of a [class], a type of flower." |
| Food101 [3] | 101 | "a photo of a [class], a type of food." |
| OxfordPets [53] | 37 | "a photo of a [class], a type of pet." |
| StanfordCars [30] | 196 | "a photo of a [class]." |
| SUN397 [75] | 397 | "a photo of a [class]." |
| UCF101 [63] | 101 | "a photo of a person doing [class]." |
| ImageNet [12] | 1000 | Ensemble of 7 selected templates, including "itap of a [class].", "a bad photo of the [class].", "a origami [class].", "a photo of the large [class].", "a [class] in a video game.", "art of the [class]." and "a photo of the small [class]." . |

## D Visualization of Graph Nodes

To demonstrate how our GraphAdapter works for the adapter-style tuning for VLMs, we visualize the graph nodes for textual features before and after the GraphAdapter. As shown in Figure 5, we randomly sampled 20 classes from ImageNet [12] and utilize the t-SNE to visualize the distribution of each node corresponding to the textual fracture for classification. We can observe that with our GraphAdapter, the nodes of different classes move in directions that lead to much larger inter-class distances, thereby improving the performance of adapter-style tuning for VLMs.

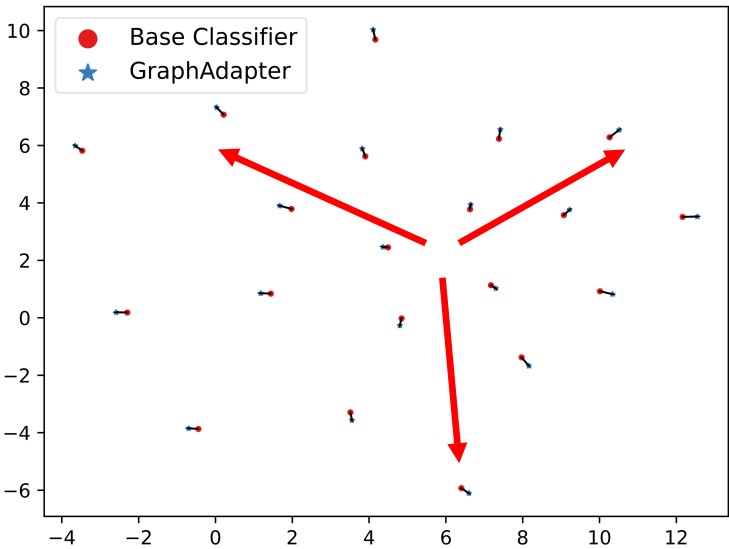

Figure 5: Visualization of the variance of the graph nodes before and after GraphAdapter. Each node represents the representation of one class. We randomly sampled 20 classes from ImageNet for better visualization. The nodes move toward the direction that leads to much larger inter-class distances after GraphAdapter. The red arrows denote the directions.

