# OpenReview forum: "GraphAdapter: Tuning Vision-Language Models With Dual Knowledge Graph"
_NeurIPS.cc/2023/Conference — NeurIPS 2023 poster_

### Official Review · Reviewer_NmuM · 2023-06-24

**Soundness:** 3 good
**Presentation:** 3 good
**Contribution:** 2 fair
**Rating:** 5
**Confidence:** 4

**Summary:**

This paper introduces GraphAdapter, a novel adapter-style tuning strategy for vision-language models. GraphAdapter can leverage task-specific structure knowledge by explicitly modeling the dual knowledge graph. The authors validate the proposed method on 11 popular benchmarks on few-shot classification setting.

**Strengths:**

-  GraphAdapter explicitly models the dual-modality structure knowledge with a dual knowledge graph, allowing for the leveraging of task-specific structure knowledge from both textual and visual modalities.
- The authors conducted comprehensive evaluations on 11 datasets and demonstrated the performance under different shots.

**Weaknesses:**

- The authors claimed the previous methods overlook the explicit exploitation of the structure knowledge, but the experimental results showed that the combination of text and visual adapters achieved limited gain (compared to text-only adapter). To some extent, it makes the motivation less convincing.
- The knowledge graph is constructed at the very beginning for once, this can be inconvenient in the setting where there will be new data coming sequentially over time, and then the knowledge graph will need to be updated every time. It's important to consider this because the reason we use adapters is we want a quick adaptation with minimal cost.
- The performance improvements seem trivial. For example, in Figure 2 and Table 1, most of the improvement over the second-best model is around 1-2%, and sometimes less than 1% or worse. It seems less promising especially considering the complicated design.


**Questions:**

Can the authors also compare the training time/parameters with previous methods to show the efficiency?

**Limitations:**

See above.

---

> ### Author Rebuttal · Authors · 2023-08-09
>
> **Q1:** The authors claimed the previous methods overlook the explicit exploitation of the structure knowledge, but the experimental results showed that the combination of text and visual adapters achieved limited gain (compared to text-only adapter). To some extent, it makes the motivation less convincing.
>
> **A1:** Thanks for your comments. We will respond in two aspects.
> (i) **An adapter typically exhibits effectiveness on a specific branch.** We claim that previous works, such CLIP-Adapter and TaskRes have shown that different types of adapters are specifically effective on a particular branch while facing performance degradation when extended to both branches.  For example, CLIP-Adapter and TaskRes are working on visual and textual branches, respectively.
> Similarly, it is reasonable that our GraphAdapter performs superior on the textual branch only.
> Moreover, it is worth noting that our method applying to two branches brings slight gains compared with other methods decreasing the performance.
> (ii)  **Our motivation**. As stated in lines 78-80, we claim that our GraphAdapter mainly captures the structure knowledge via two sub-graphs, i.e., textual sub-graph, and visual sub-graph, ***for adapting text branch only***. To show the effectiveness of exploring structure knowledge, we should compare our GraphAdapter with others that do not utilize such knowledge, instead of contrasting with variants of our own method.
> The significant performance gains against other methods substantiate our rationale for harnessing structural knowledge.
>
> **Q2:** The knowledge graph is constructed at the very beginning for once, this can be inconvenient in the setting where there will be new data coming sequentially over time, and then the knowledge graph will need to be updated every time. It's important to consider this because the reason we use adapters is we want a quick adaptation with minimal cost.
>
> **A2:** Thanks for your comments. We contend that the necessity for updating the knowledge graph hinges on the nature of incoming data.
> If the new data sharing the same label space is introduced sequentially, our graph can remain unaffected.
> In contrast, when new classes or tasks are introduced sequentially, creating a lifelong/incremental learning (IL) set-up, all adapter-style methods need re-optimization including our GraphAdapter.
> For example, TaskRes necessitates configuring a new residual, while Tip-Adapter needs cache updates and weight relearning.
> Overall, we claim that all these methods are not designed for the IL set-up.
> However, we acknowledge that the IL set-up is practical and evaluate our method under a pseudo IL setting where a model is trained on base classes and directly tested on unseen new classes.
> The results demonstrate the superiority of our method against CoOp and CoCoOp by gains 2.08\% and 0.90\%, respectively.
> Note that our method is not specifically designed for this base-to-new set-up.
> Furthermore, we provide the possibility of reformulating our method to real IL, i.e., we can introduce a dynamic graph, which would facilitate the incremental expansion of nodes and limited updating of edges correlated with the new nodes.
>
> **Q3:** The performance improvements seem trivial. For example, in Figure 2 and Table 1, most of the improvement over the second-best model is around 1-2\%, and sometimes less than 1\% or worse. It seems less promising, especially considering the complicated design.
>
> **A3:** Thanks for your thoughtful feedback. We acknowledge that the observed improvements may appear modest at first glance. However, we emphasize that while the percentage differences might seem small, they can still be meaningful for the task of tuning VLMs with few-shot samples due to the deficiency of data used for training and limited learnable parameters. For example, regarding the average accuracy over 11 diverse datasets, the number has been improved gradually across methods, e.g., 73.42\% (CoOp, IJCV) $\rightarrow$ 74.44\% (CLIP-Adapter, Arxiv2021) $\rightarrow$ 75.65\% (Tip-Adapter, ECCV2022) $\rightarrow$ 75.10\% (TaskRes, CVPR2023). This underscores the challenge of enhancing performance in this task and, consequently, highlights that the 1-2\% gains are not trivial.
>
> Moreover, while the conceptualization of constructing the dual knowledge graph to adapt VLMs might seem intricate, its implementation is remarkably straightforward and the introduced parameters are limited (only 4.145M).
> Consequently, we assert that our method is not overly complex.
>
> In summary, taking into account the ease of implementation and the modest parameter requirements that lead to 1-2\% performance gains on such a challenging task, our approach demonstrates its effectiveness and holds the potential to inspire future research.
>
> **Q4:** Can the authors also compare the training time/parameters with previous methods to show the efficiency?
>
> **A4:** Thanks for your great suggestions. In the following table, we compare our method with existing published ETL methods on the ImageNet in 16-shot case from the perspectives of tunable parameters, computational flops, training time, inference time. All results are measured with the officially released code from GitHub. We can observe our tunable parameters are still less than Tip-Adapter-F. For computational flops, our GraphAdapter takes about 5.42GFlops, almost the same as Tip-Adapter-F and TaskRes, which is far lower than CLIP-Adapter. Moreover, our training and inference times are less than the adapter-style work CLIP-Adapter and the prompt tuning method CoOp.
>
> |    | CoOp| CLIP-Adapter| Tip-Adapter-F| TaskRes| Ours|
> |:-:|:-:|:-:|:-:|:-:|:-:|
> |Tunable Parameters (M)| 0.008 | 0.524| 16.384 | 1.024| 4.145|
> |GFlops | 1943.12  | 1959.44 | 5.43 | 5.42 | 5.42|
> |Training time (one epoch) (s)| 40.91 | 45.71 | 12.36 | 13.64 | 23.29|
> | Inference time (s/100) | 119.64 | 275.22 | 51.03 | 4.89 | 4.91|
> | Performance | 62.95 | 63.59 | 65.44 | 64.75 | 65.70 |

---

> > ### Comment · Reviewer_NmuM · 2023-08-14
> > **Thanks for the rebuttal**
> >
> > The authors have addressed most of my concerns. I'm raising my score to 5.

---

> > > ### Author Response · Authors · 2023-08-15
> > > **Appreciation for Your Feedback**
> > >
> > > Thanks for your great efforts in our work. We sincerely appreciate your positive and constructive comments. We will incorporate these in our revision carefully.

---

### Official Review · Reviewer_pwW3 · 2023-06-25

**Soundness:** 3 good
**Presentation:** 3 good
**Contribution:** 2 fair
**Rating:** 5
**Confidence:** 5

**Summary:**

This paper proposes a new prompt tuning strategy named GraphAdapter to fuse textual and visual structure knowledge for downstream tasks. It first constructs the dual knowledge graph by taking the textual / visual features of a specific class as nodes and the cosine similarities between these features as edges. After that, the textual feature $z_t$ is warped into the graph space and two GCNs are applied to excavate the knowledge from the textual and visual sub-graphs respectively. Extensive experiments on 11 benchmark datasets show that GraphAdapter consistently outperforms previous works.

**Strengths:**

- The idea of introducing graph learning into prompt tuning methods is novel.

- The average performance of GraphAdapter is better than previous methods on few-shot learning, including 1-/2-/4-/8-/16-shots on 11 benchmark datasets.

- Ablations also show that GraphAdapter can achieve consistent gains when using different backbones.

**Weaknesses:**

- My major concern is about the **efficiency**. Although modeling all the classes as a knowledge graph is a novel idea, introducing GNNs into prompt tuning are generally time and memory consuming and may violate the original motivation of the *efficient* transfer learning for Vision Language foundation models (VLFMs). In fact, as is mentioned in the appendix (L40), the authors already decouple the original graph of ImageNet (1k nodes) into 4 sub-graphs (256 nodes) to alleviate the computational cost. So the *scalability* of this method maybe limited.

- Besides that, one of the most important potential of VLFMs is trasferring to *open-world* scenarios, where the classes are unknown and infinite, which may make the graph impossible to build in advance.

**Questions:**

- In the few-shot training setting, what is the detailed process to obtain the visual features for constructing the visual sub-graph? For example, what is the number of images used here? what kind of data augmentation is applied?

- what is the overall cost of GraphAdapter compared to previous methods? For example, (a) The training FLOPs; (b) The training wall-time of one epoch; (c) The inference speed; (d) The number of parameters.

- (Minor) The reference figure in L171 is incorrect.

**Limitations:**

The limitations are partly addressed. The authors may need to further discuss the limitation of **scalability** as well.

---

> ### Author Rebuttal · Authors · 2023-08-09
>
> We sincerely thank you for your great efforts and insightful questions.
>
> **Q1:** About the concern on efficiency and scalability.
>
> **A1:** Thanks for your valuable comments. We will answer the above questions from three perspectives.
>
> (i) As the definition of efficient transfer learning in VLMs, it contains two perspectives: 1) parameter-efficient transfer learning, and 2) data-efficient transfer learning. Our GraphAdapter satisfies these two principles like existing ETL methods since it only utilizes 4.145M learnable parameters and assists few-shot learning.
>
> (ii) To prove that our scheme is efficient, we compute and compare our methods with existing published efficient transfer learning methods on the ImageNet with the 16-shot setting,  from the perspectives of tunable parameters, computational flops, training time, and inference time. All results are measured with the officially released code from GitHub. The experimental results are shown in the table below. We can observe our tunable parameters are still less than Tip-Adapter-F. For computational flops, our GraphAdapter takes about 5.42GFlops, almost the same as Tip-Adapter-F and TaskRes, which is far lower than CLIP-Adapter. Moreover, our training and inference times are less than the adapter-based work CLIP-Adapter and the prompt tuning method CoOp.  Therefore, our GraphAdapter satisfies the requirement of efficient transfer learning.  Moreover, our GraphAdapter can achieve the best performance.
> |    | CoOp| CLIP-Adapter| Tip-Adapter-F| TaskRes| Ours|
> |:-:|:-:|:-:|:-:|:-:|:-:|
> |Tunable Parameters (M)| 0.008 | 0.524| 16.384 | 1.024| 4.145|
> |GFlops | 1943.12  | 1959.44 | 5.43 | 5.42 | 5.42|
> |Training time (one epoch) (s)| 40.91 | 45.71 | 12.36 | 13.64 | 23.29|
> | Inference time (s/100) | 119.64 | 275.22 | 51.03 | 4.89 | 4.91|
> Memory Cost (Training) | 18.907| 9.257 | 4.313 | 6.227 | 10.75 |
> Memory Cost (Inference) | 7.403 | 7.615 | 4.161 | 6.225 | 4.433 |
> | Performance | 62.95 | 63.59 | 65.44 | 64.75 | 65.70 |
>
> (iii) For adapter-style efficient transfer learning methods in VLMs, the computational complexity will increase when the number of classes increases in the training process. For example, the complexity of the adapter process in CLIP-Adapter, Tip-Adapter-F, and TaskRes are all $O(kn)$, where $k$ is constant. In our GrapAdapter, **the number of parameters of GCN is invariant for any class number.** The increase in computational cost is primarily caused by edge computing. To reduce the complexity, we decompose the 1000 nodes in ImageNet into four 250 nodes for the trade-off of performance and computational complexity. Here, we give the theoretical derivation. If we have n class, we can divide it into several subgraphs with fixed $m$ nodes, and decrease the edge computing to $O(mn)\sim O(n)$ when n is large since $m$ is constant like $k$. We also conduct experiments for different $m$ as the table below. We can find with the decrease of $m$, the performance only drops a little.
> Therefore, it owns almost the same scalability as existing adapter-style ETL methods, like TaskRes, CLIP-Adapter, and Tip-Adapter-F.
> | m | 20  | 50 | 100 | 250 |
> |:-:|:-:|:-:|:-:|:-:|
> |1-shot | 61.23 | 61.45 |61.47| 61.50|
>
> **Q2:** One of the most important potentials of VLFMs is transferring to open-world scenarios, where the classes are unknown and infinite, which may make the graph impossible to build in advance.
>
> **A2:** Thanks for your insightful and valuable comments. (i) First, our GraphAdapter follows the CoOp setting, which is followed by existing adapter-style works and is devoted to improving the transferability to the seen task, including seen classes or domain generalization scenarios. Therefore, it is not specifically optimized for open-world scenarios, the same as existing adapter-style ETL works and CoOp.
> (ii) But, we find that our GraphAdapter inherently owns the transferability for open-world scenarios despite no special optimization or design. Notably, **in our GraphAdapter, once the training is finished, the dual knowledge graphs of our GraphAdapter are stored in models, which are not required to be constructed in the inference stage in advance even for open-world scenarios.** As shown in Fig. 2 of our manuscript, given the novel classes, their textual features can warp knowledge from an existing dual knowledge graph constructed with few-shot training data, for the classification of novel classes. CoCoOp is an excellent work for the trade-off between seen classes and unseen classes in an open-source scenario. We follow the base-to-new setting in CoCoOp, where the new classes are unseen in the training process. The experimental results are shown in the Table below:
> Methods | Base  | New | H
> |:-|:-:|:-:|:-:|
> CLIP  | 72.43  | 68.14 | 70.22
> CoOp  | 76.47  | 67.88  | 71.92
>  CoCoOp | 75.98 | 70.43 |  73.10
>  GraphAdapter (Ours) |  77.91 | 70.13 | 74.02
>
> We can find our GraphAdapter can achieve a better harmonic mean (higher generalization trades-off than CoCoOp). And our GraphAdapter can achieve 70.13\% accuracy for new classes, which outperforms CoOp by a large margin.
>
> **Q3:** What is the detailed process to obtain the visual features for constructing the visual sub-graph?
>
> **A3:** The visual sub-graph is constructed with the few-shot training samples. Before the training, we exploit the visual encoder of pre-trained CLIP to extract the visual features for these few-shot samples from the same class and average them as the node of this class. The number of images used for each class is decided by the number of shots. For example, for the 2-shot task, two images are used for each node. The data augmentation strategy only contains "random resized cropping" and "random flipping". We will add this description in our revision.
>
> **Q4:**  What is the overall cost of GraphAdapter **A4:**  Please see the A1.
>
> **Q5:** (Minor) The reference figure in L171 is incorrect. **A5:** Thanks for it. We will revise it in the revision carefully.

---

> > ### Author Response · Authors · 2023-08-18
> > **Sincerely expect your response**
> >
> > Dear reviewer pwW3:
> >
> > Thank you for your great efforts in reviewing our paper and providing constructive suggestions/comments. If our rebuttal does not address your concerns, you are warmly welcomed to raise questions.
> >
> > Best Wishes!
> >
> > Authors

---

> > > ### Comment · Reviewer_pwW3 · 2023-08-20
> > >
> > > Thanks for the author's response, most of my questions are addressed. I will raise my original rating to 5 (borderline accept).

---

> > > > ### Author Response · Authors · 2023-08-20
> > > > **Appreciate for your response**
> > > >
> > > > We are glad that you are satisfied with our rebuttal. Thanks for your insightful comments and your suggestions will be incorporated in our revision. Thank you once again for your constructive review.

---

### Official Review · Reviewer_hFHN · 2023-07-01

**Soundness:** 3 good
**Presentation:** 4 excellent
**Contribution:** 3 good
**Rating:** 7
**Confidence:** 4

**Summary:**

The paper proposes to utilize graph learning for efficient transfer learning of large vision-language models. The graph learning consists of scene graphs of two different modalities, first one uses the textual features of prompts and the second one uses the visual features of training samples from the downstream tasks. The feature of each prompt aims to align with the the fused graph features during the optimization process. Experiments on 11 few shot classification benchmarks including a few on fine-grained classification tasks show that this method improves the transfer learning performance.


**Strengths:**

The idea of using structured knowledge is intuitive. Using textual and visual both modalities are interesting. The section of different graph adapter variants, text, visual and T-V is worth adding. The ablation experiments of the different coefficients for different modality graphs have added value. The paper proposes an intuitive idea, explains it well and the paper is well written.

**Weaknesses:**

Not a weakness, but a general question.
Is there a way to utilize the semantics of visual relationships? The structured knowledge graph in its current form appear to be more of a co-occurrence statistics.

**Questions:**

This is a well written paper. I have one question though:
Is there a way to utilize the semantics of visual relationships? The structured knowledge graph in its current form appear to be more of a co-occurrence statistics. e.g. How to distinguish between "person holding a cup" vs "person drinking from a cup" ?

**Limitations:**

Broader impacts and limitations are mentioned in the supplementary material, it's the common impact that large vision-language models need to be concerned about - the nature of the pretraining dataset.

---

> ### Author Rebuttal · Authors · 2023-08-09
>
> Thanks for your recognition of our work. We have given careful consideration in response to your insightful question.
>
> **Q1:** Is there a way to utilize the semantics of visual relationships? The structured knowledge graph in its current form appears to be more of a co-occurrence statistic. e.g. How to distinguish between "person holding a cup" vs "person drinking from a cup"?
>
> **A1:** Thanks for your positive comments and insightful questions. Our GraphAdapter, following the setting of CoOp, has been validated on 11 downstream benchmarks for classifications. We believe the semantics of the visual relationships you mentioned can further enhance classification accuracy. Intuitively, the utilization of the visual relationship should be particularly beneficial for fine-grained and multiple-object classification in one image.
>
> Based on your insightful question, we conduct a survey of related works and found that two particular directions are closely correlated with the visual relationship within a single image, such as distinguishing  "person holding a cup"
> and "person drinking from a cup". These two directions compose of human-object interaction[1, 2, 3] (HOI) and scene graph generation[4, 5] (SGG). Among them, human-object interaction focuses on identifying the relationship between humans and nearby objects.  Scene graph generation is designed to identify the relationship between different objects within a single image. Both of these two tasks typically involve first detecting the objects/humans, followed by classifying their relationship.
>
> The above directions inspire us to consider two potential approaches to exploit the visual relationship: i) utilizing paired visual images and annotated language description of the relationship to guide both the CLIP textual classifier and the visual encoder, enabling them to be aware of the relationship. ii) detecting the objects within an image and modeling the relationship through graph learning as the SGG works.
>
> Two potential challenges arise in the utilization of visual relationships.
>  i) Whether the annotated datasets for visual relationships are adequate for classification
>  ii) how to design one efficient and effective scheme to exploit the visual relationships for classification.
>
> We believe it is a very good direction to investigate in future work.
>
> [1] Gkioxari, Georgia, Ross Girshick, Piotr Dollár, and Kaiming He. "Detecting and recognizing human-object interactions." In Proceedings of the IEEE conference on computer vision and pattern recognition, pp. 8359-8367. 2018.
>
> [2] Liao, Yue, Aixi Zhang, Miao Lu, Yongliang Wang, Xiaobo Li, and Si Liu. "Gen-vlkt: Simplify association and enhance interaction understanding for hoi detection." In Proceedings of the IEEE/CVF Conference on Computer Vision and Pattern Recognition, pp. 20123-20132. 2022.
>
> [3] Park, Jeeseung, Jin-Woo Park, and Jong-Seok Lee. "ViPLO: Vision Transformer based Pose-Conditioned Self-Loop Graph for Human-Object Interaction Detection." In Proceedings of the IEEE/CVF Conference on Computer Vision and Pattern Recognition, pp. 17152-17162. 2023.
>
> [4] Lin, Xin, Changxing Ding, Yibing Zhan, Zijian Li, and Dacheng Tao. "Hl-net: Heterophily learning network for scene graph generation." In proceedings of the IEEE/CVF conference on computer vision and pattern recognition, pp. 19476-19485. 2022.
>
> [5] Tang, Kaihua, Yulei Niu, Jianqiang Huang, Jiaxin Shi, and Hanwang Zhang. "Unbiased scene graph generation from biased training." In Proceedings of the IEEE/CVF conference on computer vision and pattern recognition, pp. 3716-3725. 2020.

---

> > ### Comment · Reviewer_hFHN · 2023-08-16
> > **post-rebuttal comments**
> >
> > I thank the authors for posting the rebuttals. I'll keep my original "accept" rating

---

> > > ### Author Response · Authors · 2023-08-20
> > > **Thanks for your positive comments**
> > >
> > > We are greatly appreciated for your support and great suggestions for our work. We believe these insightful comments can bring one potential direction for the tuning of VLMs.

---

### Official Review · Reviewer_hUWw · 2023-07-05

**Soundness:** 4 excellent
**Presentation:** 4 excellent
**Contribution:** 4 excellent
**Rating:** 8
**Confidence:** 5

**Summary:**

In this paper, the authors present an adapter-style tuning method, termed as GraphAdapter, that explicitly captures the dual-modality structure knowledge by utilizing a dual knowledge graph, leading to enhanced adapter-style transfer learning. Specifically, the authors identify two key challenges in existing adapter-style approaches for efficient transfer learning, including the deficiency in modeling task-specific knowledge from only a single modality, and the neglect of explicitly exploiting the structural knowledge in downstream tasks. Motivated by these two limitations, the authors propose a novel tuning method for visual-language models that incorporates task-specific knowledge for downstream tasks through the integration of textual and visual structure knowledge, based on graph learning. In particular, the proposed method establishes a dual knowledge graph consisting of a textual knowledge subgraph and a visual knowledge subgraph. Consequently, the feature adapter can effectively leverage the inner-modality and cross-modality structure knowledge for superior tuning performance. Experiments on 11 benchmarks convincingly demonstrate the effectiveness of the proposed GraphAdapter approach.

**Strengths:**

1. The proposed GraphAdapter is very well-motivated. The authors have conducted a thorough analysis of the existing ETL approaches, identifying two key limitations, and recognizing the significance of incorporating dual-modality structure knowledge in ETL. Building upon such analysis, the authors develop GraphAdapter, which aims to effectively integrate fused textual and visual structure knowledge using GCN.

2. The experiments are thorough and convincing. The authors perform extensive experiments on 11 few-shot benchmarks, utilizing various backbones such as ResNet-50, ResNet-101, ViT-B/32, and ViT-B/16, as detailed in both the main paper and the supplementary material. In addition, the authors have specifically explored the generalization capability of GraphAdapter on four benchmarks. These experiments convincingly demonstrate the effectiveness of GraphAdapter.

3. The paper is clearly written and easy to follow. The authors extensively elaborate on the details of GraphAdapter, particularly the establishment of the dual knowledge graph.

4. The supplementary material includes extensive implementation details and visualization results. Notably, the authors demonstrate the seamless integration of the proposed GraphAdapter with existing methods such as CaFo and TaskRes*, resulting in consistently improved performance.

**Weaknesses:**

I’m almost satisfied with this paper, with only a few minor concerns as follows.

1. The authors leverage GCN to integrate dual-modality structural knowledge. However, I am interested in understanding the performance of more advanced GNN mechanisms, such as GAT and GraphSAGE, in this context.

2. The authors are suggested to include an analysis of the time complexity, particularly regarding the construction of the textual and visual knowledge subgraphs.

3. Some related works [1, 2, 3] on KG embedding can be included.

[1] Wu, Han, et al. "Hierarchical Relational Learning for Few-Shot Knowledge Graph Completion." The Eleventh International Conference on Learning Representations. 2022.
[2] Bordes, Antoine, et al. "Translating embeddings for modeling multi-relational data." Advances in neural information processing systems 26 (2013).
[3] Xiong, Wenhan, et al. "One-shot relational learning for knowledge graphs." arXiv preprint arXiv:1808.09040 (2018).

**Questions:**

1. Can GCN in GraphAdapter be replaced with other GNNs like GAT and GraphSAGE?
2. What is the time complexity associated with GraphAdapter?

**Limitations:**

The limitations have been thoroughly discussed in the main paper.

---

> ### Author Rebuttal · Authors · 2023-08-09
>
> We greatly appreciate your positive comment on our work, along with constructive suggestions for the improvement of our work.
>
> **Q1:** The authors leverage GCN to integrate dual-modality structural knowledge. However, I am interested in understanding the performance of more advanced GNN mechanisms, such as GAT and GraphSAGE, in this context.
>
> **A1:** Thanks for your valuable comments. In order to delve further into this intriguing question, we replace the GCN in our GraphAdapter with GAT and GraphSAGE. Then we compare them on the Caltech dataset using a 4-shot setting.
> As shown in the table below, the utilization of a more advanced graph neural network (GNN) mechanism (such as GAT and GraphSAGE) result in a slight improvement in performance. Meanwhile, it also brings more resource const. We will add this analysis in the revision.
>
> Methods| GAT | GraphSAGE | GCN |
> |:-:|:-:|:-:|:-:|
> |Caltech|91.21| 91.19| 90.97|
>
> **Q2:** The authors are suggested to include an analysis of the time complexity, particularly regarding the construction of the textual and visual knowledge subgraphs.
>
> **A2:** Thank you for your insightful suggestions. In response, we have incorporated an analysis of the time complexity for our GraphAdapter in the table below. This includes the training time for each epoch, the inference time for a batch of 100 images, and the associated costs for constructing both the textual and visual graphs. All the time complexity computations were performed with the ImageNet dataset on a 16-shot setting, using a single NVIDIA GeForce 3090. As shown in the table below, the time required for training and inference of our GraphAdapter is obviously less than the typical prompt-based ETL method CoOp and Adapter-based work CLIP-Adapter, which is efficient. Moreover, the textual and visual graphs need only be constructed once at the beginning of the training process. It means the construction of these two graphs is also fast and the cost is minimal.
> |Methods| Training time (one epoch) | Inference time | Textual Graph |Visual Graph| Performance|
> |:-|:-:|:-:|:-:|:-:|:-:|
> |CoOp| 40.91s | 119.64s | -  | - | 62.95 |
> |CLIP-Adapter | 45.71s | 275.22 | - | - | 63.59|
> |Ours| 23.28s| 4.57s | 6.63s | 14.34s| 65.70|
>
> **Q3:** Some related works [1, 2, 3] on KG embedding can be included.
>
> **A3:** Thank you for your valuable suggestions. We will incorporate descriptions of these papers in the "Graph Learning" section of our related work.

---

> > ### Comment · Reviewer_hUWw · 2023-08-20
> >
> > Thank you for the response. The response addressed most of my concerns. Assuming the response will be incorporated to final manuscript, I raise my vote further. I strongly support acceptance for this paper.

---

> > > ### Author Response · Authors · 2023-08-20
> > > **Appreciate for your response**
> > >
> > > Thank you for your response and strong support for our paper. Following your valuable suggestions. we assure you that the points raised in the response will indeed be incorporated into the final manuscript.  We are grateful for your encouragement, and your constructive comments have greatly helped us to further improve our work.

---

### Author Rebuttal · Authors · 2023-08-10

**We thank all reviewers and area chairs for their great efforts and insightful comments!** These suggestions and questions are significantly beneficial to our paper. We believe we have addressed all the concerns of reviewers in the rebuttal. **If you have some new questions/concerns, please let us know.** We will try our best efforts to solve it. Thanks.

---

### Decision · Program_Chairs · 2023-09-21

**Decision:**

Accept (poster)

**Comment:**

Reviewers have reached a positive consensus on this work. The concerns of reviewers are well addressed; the contributions and the effectiveness are supported by reviewers. Overall, it is recommended to be accepted.